# Vascularized Growth Plate Transfer in Paediatric Ulna Non-Union: Operative Technique and Review of the Literature

**DOI:** 10.3390/jcm12154981

**Published:** 2023-07-28

**Authors:** Nisha M. Grünberger, Amelie Klein, Marina Barandun, Dirk J. Schaefer, Andreas H. Krieg, Alexandre Kaempfen

**Affiliations:** 1Department of Plastic, Reconstructive, Aesthetic and Hand Surgery, Basel University Hospital, University of Basel, 4001 Basel, Switzerland; nishamercedes.gruenberger@usb.ch (N.M.G.); amelie.klein@hcuge.ch (A.K.); m.barandun@belcare.ch (M.B.); dirk.schaefer@usb.ch (D.J.S.); 2Paediatric Orthopaedic Department, University Children’s Hospital, 4031 Basel, Switzerland; andreas.krieg@ukbb.ch

**Keywords:** vascularized growth plate transfer, neurofibromatosis type 1 (NF1), congenital pseudoarthrosis forearm, pediatric hand

## Abstract

Congenital pseudarthrosis of forearm fractures is rare and is strongly associated with neurofibromatosis type 1 (NF1). Our case report illustrates the progression of a non-union of the ulna after minor trauma in a twelve-year-old boy, newly diagnosed with NF1, and presents the technique of microsurgical bone reconstruction, including the growth plate. More than seven years after the first operation, follow-up presents a favorable outcome with a pain-free patient and unrestricted function of the forearm after a secondary correction of the remaining radial bowing. This treatment is discussed with a comprehensive review of the current literature on ulnar congenital pseudarthrosis in PubMed and Google Scholar and free fibular growth plate transfer in PubMed and Google Scholar. Nine publications reporting on 20 cases of congenital ulnar non-unions were identified. With this reconstructive option, favorable outcomes were achieved in all cases with the union after primary surgery and complications requiring further surgeries in nine cases. The benefit of vascularized growth plate bone transfer in congenital ulna non-union seems to be significant compared to other therapies such as open reduction internal fixation (ORIF), non-vascularized bone grafts, or one-bone-forearms and beneficial when growth reconstruction is needed. Other techniques might be necessary to improve insufficient long-term results.

## 1. Introduction

Congenital pseudarthrosis of the ulna and radius are rare conditions of altered bone growth and callus formation. They are often present with pathologic fractures and the development of longstanding pseudarthroses due to abnormal mechanisms of fracture healing [1]. Previous studies illustrated the high association with neurofibromatosis type 1 (NF1). In fact, 40% to 73% of patients presenting with congenital pseudarthroses of a forearm bone suffer from neurofibromatosis [1,2,3,4]. According to Ding et al., in 2017, this hereditary disease worldwide affected one in 3000 live births [5]. Until recently, only 68 cases of forearm pseudarthrosis have been reported in English literature, 29 of which had been isolated ulnar cases, 19 radius cases, and in 20 cases, both bones were affected simultaneously [4,6,7,8]. Charles et al. [4] reported 50% pseudarthrosis of the ulna, 26% of the radius and 24% affected both forearm bones. The distal half is most commonly affected in an ulnar pseudarthrosis.

Pseudarthrosis of the ulna may cause growth disturbance and progressive forearm deformity, which can cause pain and functional impairment of forearm motion—especially rotatory. Treatment is still challenging, and there should be three goals included in surgical decision-making: bone healing, distal radioulnar joint (DRUJ) stability and motion, and skeletal growth. Cast immobilization, corrective osteotomies, internal fixation with intramedullary nail or plate and screw constructions, and non-vascularized bone grafting have been used to treat this condition but often resulted in unsuccessful bone healing or did not achieve sufficient long-lasting deformity correction [1,9,10,11]. Free vascularized fibula transfer, first described by Allieu et al. [12], gained popularity over time. Specifically, microsurgical advances made this procedure more reliable and provided superior results compared to other techniques. A substantial contribution to this evolution was made by Innocenti et al. [13,14], who provided a clear description of the flap dissection of the fibula, including the growth plate, thus enabling the reconstruction of growth reliably. However, only a few cases of ulnar congenital pseudarthrosis have been treated with a free vascularized fibular growth plate (FVFGP) transfer, and these cases have never been comprehensively and systematically reviewed [1,6,7,10,15,16,17,18] regarding outcomes. Therefore, the aim of this work is to present the rationale and reliability of our treatment regimen for a young patient suffering from congenital ulna non-union and to discuss this with a review of the existing literature.

## 2. Case Report

### 2.1. Case Report

A twelve-year-old boy was admitted to an external hospital due to a fracture of the right distal ulna after a football accident. The initial treatment with an upper arm cast for five months ended in pseudarthrosis (Figure 1a,b), with ulnar deviation at the forearm. However, there was no alteration of wrist, finger, or elbow range of motion at this stage.

The main complaint of the patient was the disfigurement; pain was secondary. The clinical examination at our children’s hospital revealed a dozen café au lait spots that pointed to the differential diagnosis of neurofibromatosis type 1, which was subsequently confirmed by genetic testing. Clinical history confirmed a trauma-independent longstanding deformity of the forearm. Old family photographs proved a long-lasting problem with a curved forearm matching the diagnosis of congenital non-union. Radiographs showed the non-union in the distal third of the right ulna, with fading and sclerosis of the long bone segment (Figure 2). Furthermore, due to ulnar shortening, the radius was bent towards the ulna, as expected from the photographs.

Nine months after the accident, the radial curvature increased to 30°, and the patient presented with limited pronation of 45° and pain on palpation of the non-union site. The MRI confirmed the increased bone remodeling and altered bone healing (Figure 3).

Pain and worsening of the deformity warranted bone reconstruction. The challenging length of the defect needed a vascularized bone reconstruction, which was only possible with a free vascularized fibula transfer. Intraoperatively, loupe magnification, clinical resistance of the bone, and histology revealed hypoperfusion and lack of stability, including the distal head of the ulna, including the epiphysis. For bony growth preservation, the fibula was microsurgically transferred, including the proximal growth plate, as described by Innocenti et al. [14]. All nerves, from the deep peroneal nerve to the anterior tibial muscle and the extensor hallucis, were meticulously dissected and preserved. The proximal bone was fixed with a plate and screws, and distally, the triangular fibrocartilage complex (TFCC) was reinserted by a transcartilaginear suture into the proximal fibula. The interosseous membrane was secured to the periosteum on the fibula. DRUJ stability was checked with a ballottement test in neutral and full pro- and supination. The arterial microsurgical anastomosis was performed with an end-to-side technique to the ulnar artery in the proximal forearm, whereby a reverse flow was achieved. At three months follow-up, the graft showed good integration and bony healing, and after six months, the patient was able to attain a normal range of motion. The graft continued to grow until skeletal maturity; however, radial curvature persisted and was not corrected despite the growing ulna. The patient was able to restart soccer training three months post-operatively. No weakness in toe or ankle extension was noted. However, because of persistent curvature of the radius, a correction was desired by the patient 6 years after the original reconstruction and after the growth of skeletal maturity. For the prevention of an excellent range of motion, a computer-assisted radial osteotomy was necessary. Preoperative virtual planning and customized 3D-printed cutting guides ensured a predictable outcome. (Figure 4) (Medacta AG international www.mysolution.medacta.com/CARD Balgrist www.balgriststiftung.ch).

Six weeks after radial osteotomy, the patient was pain-free and presented a supination of 90° and pronation of 60–70°—similar to preoperative measurements after reconstruction, to the contralateral side and according to the planning. Radiographs showed the correct position of the implants with initiated bony healing (Figure 5).

At follow-up two years after radial osteotomy and one year after plate removal, no significant functional deficiency of wrist motion was noted. Furthermore, the patient was still pain-free and was pleased with the cosmetic upgrade (Figure 6a–d).

### 2.2. Literature Review

#### 2.2.1. Material and Methods 

Due to our case, we decided to analyze the existing literature and reviewed only English literature describing the FVFGP transfer to treat congenital ulnar non-union, which was performed until April 2023 to evaluate the efficiency of the procedure as demonstrated by bone consolidation and postoperative complications of the procedure.

The search was performed in PubMed and Google Scholar, and the following search term was used: “congenital pseudoarthrosis ulna” and “free vascularized growth plate transfer”. The references of the thereby identified publications were screened for additional matching manuscripts. All relevant papers were manually screened and reviewed to extract the data according to the predetermined criteria. The following information was documented and tabulated for each article: 1. year of publication, 2. name of first author, 3. number of cases, 4. age of patient, 5. diagnosis of NF 1, 6. follow-up duration, 7. complications, 8. need for further surgeries, and 9. outcomes in terms of union/non-union. We excluded literature reviews and clinical studies in which the FVFGP transfer was performed for a radial non-union or non-union of both forearm bones, in which adult patients were included, and in which the outcomes regarding bony union or graft survival were reported imprecisely. Data is presented descriptive only, as a statistical analysis was impossible for this small series of reported patient outcomes.

#### 2.2.2. Results

Nine articles published between 1993 and 2021 matched our inclusion criteria. No articles were excluded. Twenty-one treated cases, with a postoperative follow-up ranging between 8 months and 17 years, were documented. Final osseous consolidation was reported for all 21 patients. 11 of these patients reached union after primary operation, without further surgery. For the remaining 10 patients, a second procedure or more was necessary; however, in total, all of them reached union, see Table 1. Four patients needed an additional cancellous bone graft, and two were treated with hardware removal [1,7,16,19]. Three patients experienced complications during growth and needed further interventions. Due to slow growth in the ulna, three patients developed a radial head subluxation or radial bowing of the forearm, all of which could be treated with an osteotomy [1,19,20]. Details concerning complications and further surgeries are reported in Table 1. To be precise, 5 of 10 case reports described non-union, especially of the proximal junction of the fibula graft or on the distal part. Bowing of the radius, shortening of the ulna, and cross-union between both bones were other issues. Furthermore, delayed union, symptomatic hardware, and fibula wound drainage were reported [1,7,10,17,18]. Due to these complications, further surgeries were necessary, whereby iliac bone grafting and bone stabilization, bone fixation and bone lengthening, callus removal between both bones, posterolateral dislocation of the radial head, and osteotomy with or without bone grafting were presented [1,7,10,12,17,18]. All of the patients presented a union in the follow-up. Our patient went on to direct union with preserved growth but needed radius correction without other complications.

As presented in Table 2 ROM seemed impaired in all cases, except in ours. We report normal wrist and elbow as well as Pronation and Supination.

## 3. Discussion

Congenital pseudoarthrosis, especially of the forearm, is a rare condition, and it is most commonly associated with NF1, whereby isolated pseudoarthrosis of the ulna is the most common deformity, followed by pseudarthrosis of both the radius and ulna [21]. Siebelt et al. [21] described that 74% of the patients with pseudoarthrosis were diagnosed with NF1. However, it is believed that this number is underestimated because of insufficient testing or unreported test results. Although NF 1 usually presents later after birth, Elfattairy et al. [22] presented a case of a newborn male with congenital pseudoarthrosis of radius and ulna that manifested at birth, with diagnosed NF 1 shortly after birth. It has to be mentioned that the rate of non-union of NF1-associated congenital pseudoarthrosis is significantly higher, with a percentage around 44–73% compared to non-NF1-associated disease (0–45%), which makes an early diagnostic approach important [9,23]. Due to the rare condition of the forearm, there is no treatment guideline, which makes the decision-making more challenging for the team in charge, and each case has to be treated individually [22].

Historically, various treatment modalities have been applied in patients with non-union in all long bones (e.g., radius, ulna, and tibia). Prolonged immobilization in a cast, corrective osteotomy, internal rodding, plating, and inductive membrane techniques [18,24] are used. With these techniques, results are poor in long-term outcomes, and bony healing is rare in congenital non-union. In 1975, Taylor et al. [24] reported the first microvascular reconstruction of a tibial non-union by using a vascularized fibula transfer. Allieu et al. [12] described this technique for a forearm non-union reconstruction. This technique became even more practicable after the improvement in the flap harvesting technique that was initially proposed by Langenskiöld [25] and then perfected by Bauer et al. [7], which is associated with minimized donor site complications. After around 20 years in use, Witoonhart et al. [10] reported excellent outcomes of the first 17 cases of congenital pseudarthrosis of the forearm (radius, ulna, and both bones) treated by free vascularized fibular graft in 1999. They proposed to perform this procedure earlier in cases of congenital long bone non-union. Ding et al. [5] reported the first case of double barrel reconstruction of a forearm bone due to the thin diameter of the donor fibula (8 mm). If the epiphysis is close to the pseudarthrosis, growth is impaired and has to be reconstructed in young children.

Our review of the literature demonstrated that FVFGP transfers are mainly used to reach the following three goals: 1. bony healing, 2. articular stability, and 3. preservation of growth [4,16,18,19,26]. In comparison to classical cancellous bone graft techniques, it provides two important advantages: it allows for a reconstruction of long segmental bone defects, which means that a more radical debridement of the pseudoarthrosis is permitted [27]. This, in theory, should allow for better bone healing in these challenging cases, which was confirmed in our series with a primary healing rate of 54%.

Additionally, just 3 of 22 cases suffered from instability, which was treated sufficiently by osteotomy.

However, the indication for this technique requires cautious review and has to be discussed carefully with the patient and parents. Especially due to the low case numbers, the long-term outcome is not estimable and has to be considered. In our review, a high number of revision surgeries and complications was presented (5 of 9 reports). Additionally, in our case, a corrective osteotomy was necessary in the long term, as fibular growth was not able to correct the initially bent radius.

FVFGP transfer requires a potentially difficult microsurgery. Donor-side morbidity with ankle instability, residual ankle pain, and random hypertrophy of the fibula are described [28,29]. In our case, we were able to preserve the extension of the hallux and ankle joint due to the meticulous preservation of the peroneal nerves. A detailed description of the harvesting technique is readily available today due to the work of Innocenti et al. [13,14]. This has probably led to a decrease in revision surgery, which is noted in the literature review over the decades.

For Charles et al. [4,26], worsening of deviation and loss of function represent appropriate indications for an FVFGP transfer. Regarding the timing of the intervention, Cheng et al. [7,15] recommend an early FVFGP. Indeed, the main problem of isolated ulna pseudoarthrosis is an imbalance of longitudinal radial and ulnar growth. Due to ligament connections, bending of the radius and, ultimately, dislocation at the radio-humeral joint with loss of pro- and supination can occur if the procedure is delayed [1,7]. If the proximal radius presents dislocation, the indication for ulnar reconstruction is controversial and discussed. Indeed, our literature review has revealed that in case of late presentation with radial head dislocation, transport of the radius by distraction of the pseudarthrosis of the ulna and reconstruction by FVFGP or a one-bone forearm may be beneficial. In our case, luckily, this was not necessary, as the radial head was not dislocated. Additionally, by planning the corrective osteotomy on the radius virtually, a prediction of pronosupination postoperative was possible and achieved.

Our case illustrates several aspects of the historical evolution of the reconstruction of congenital ulna non-union. At first, a conservative treatment failed and could have been avoided if a correct diagnosis had been made earlier, which is a common issue as NF presents later after birth. Second, a reconstruction of the growth plate of the ulna with the fibular epiphysis is feasible and today follows standardized procedure guidelines that minimize complications and maximize outcomes. Third, the growth of the fibula matched that of a healthy ulna, and already established deformity cannot be normalized by growth.

## Figures and Tables

**Figure 1 jcm-12-04981-f001:**
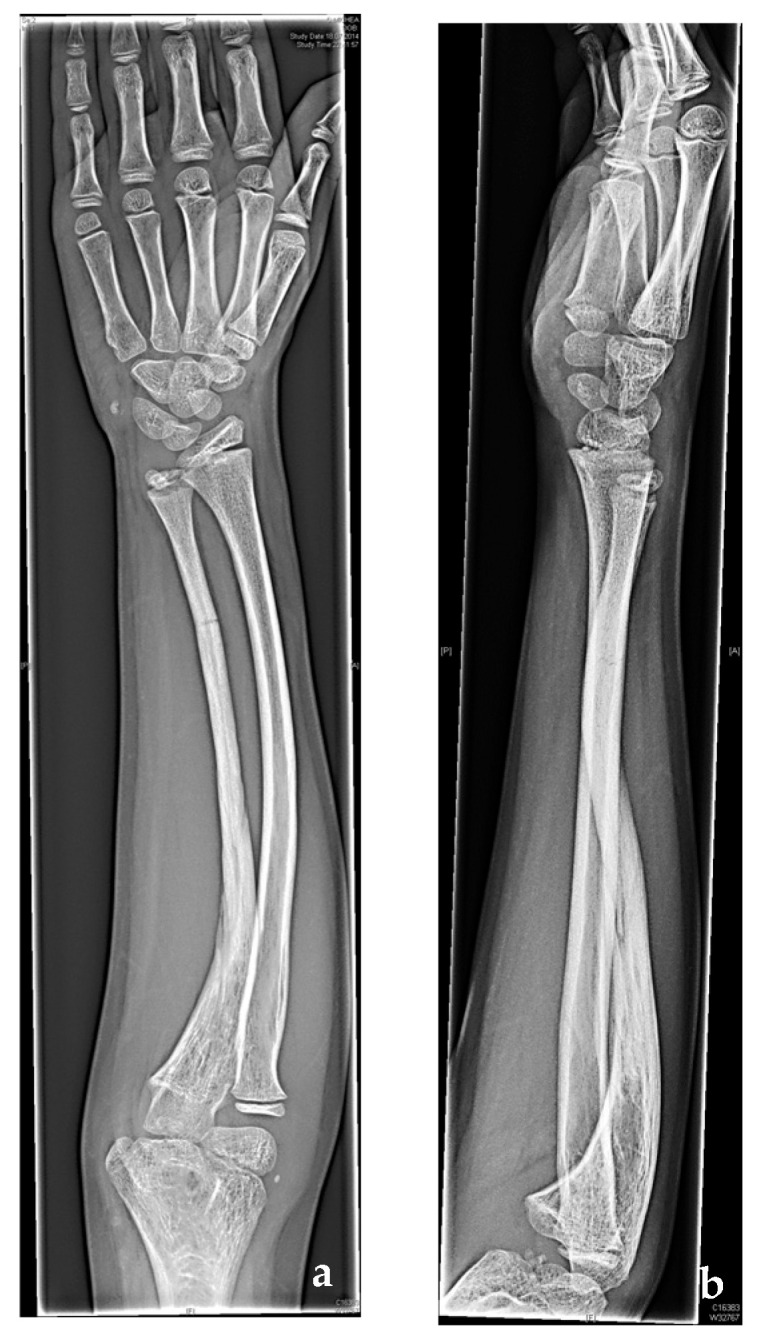
(**a**) Anteroposterior radiograph and (**b**) lateral radiograph of the non-union of the right ulna directly after minor trauma at the local hospital.

**Figure 2 jcm-12-04981-f002:**
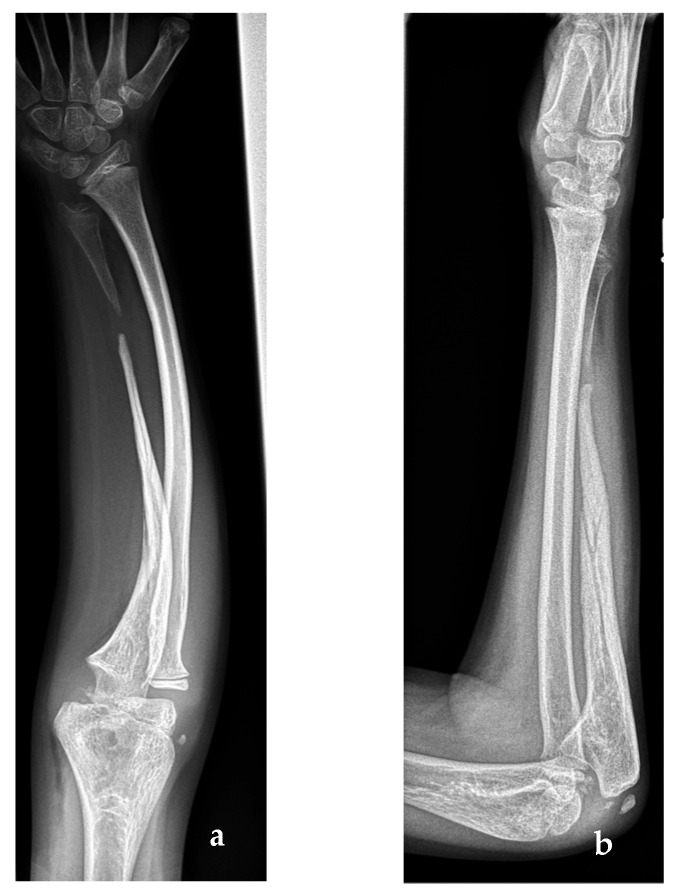
(**a**) Anteroposterior radiograph and (**b**) lateral radiograph of the established pseudoarthrosis of the right ulna at first presentation in the specialist clinic 6 months after the first X-ray.

**Figure 3 jcm-12-04981-f003:**
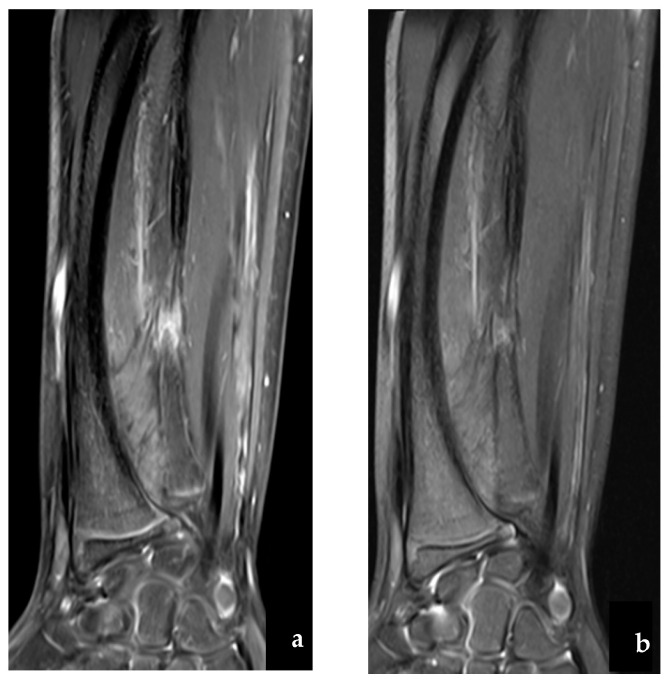
(**a**) MRI confirming the pseudoarthrosis of the right ulna in T1 and (**b**) in T2-weighted images.

**Figure 4 jcm-12-04981-f004:**
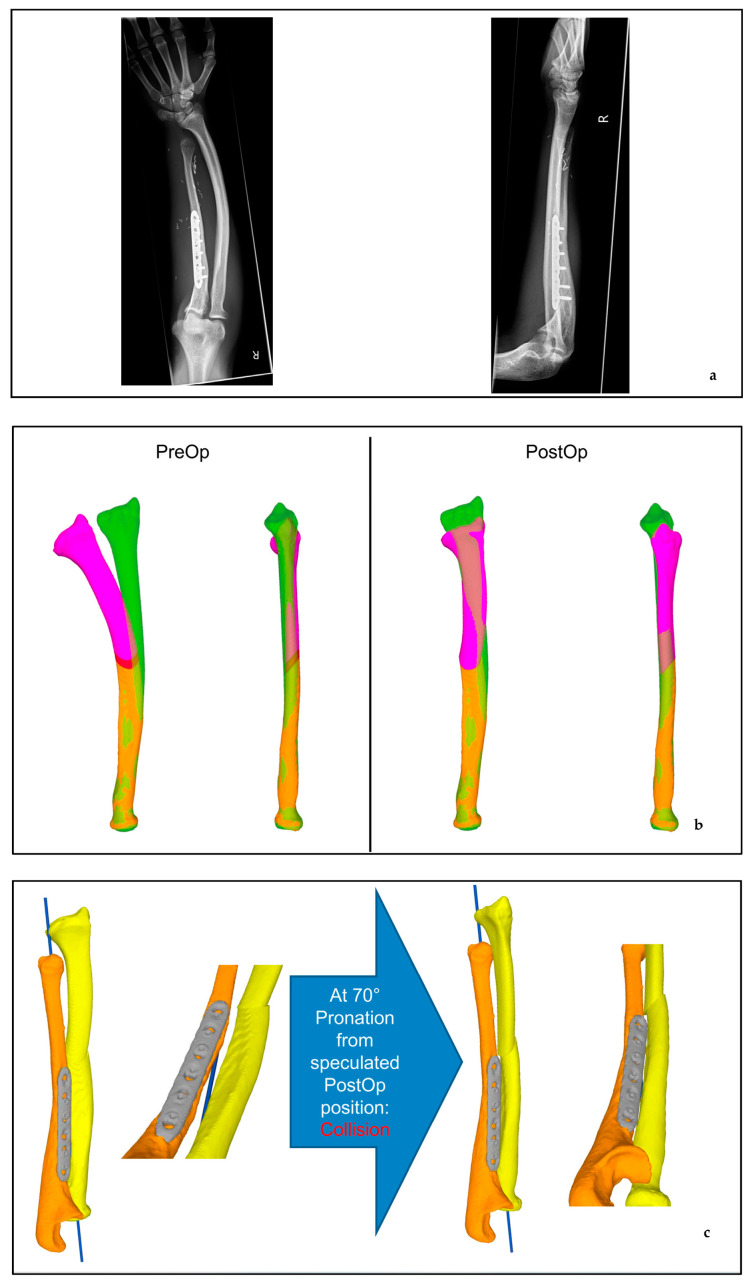
(**a**) X-rays before osteotomy (**b**) 3D deformity analysis before Operation (PreOP) versus predicted postoperative (PostOP) pink = mobilized distal fragment, green = mirrored contralateral (**c**) predicted collision.

**Figure 5 jcm-12-04981-f005:**
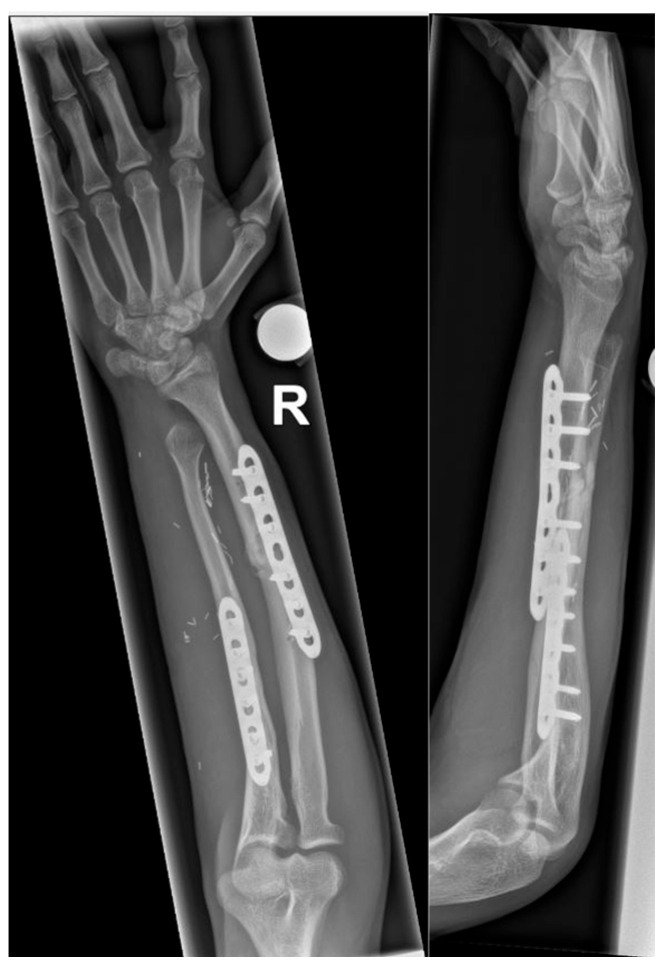
Anteroposterior and lateral view six weeks after radial osteotomy.

**Figure 6 jcm-12-04981-f006:**
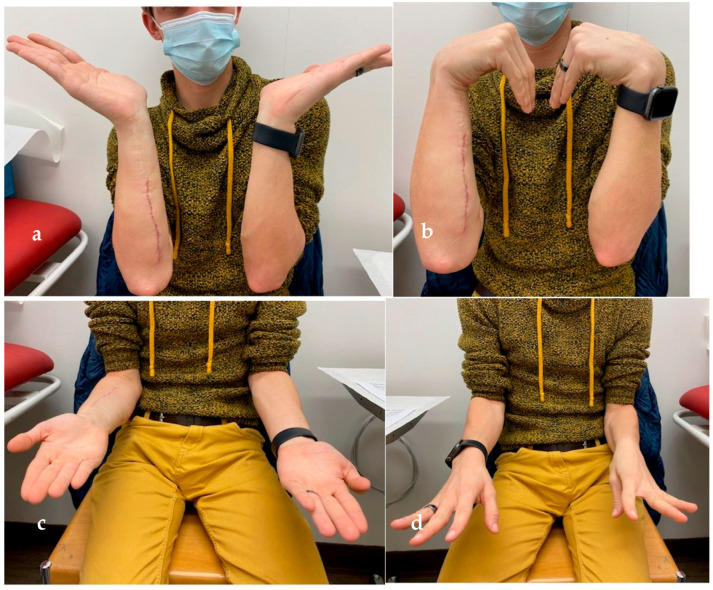
(**a**) Almost normal wrist extension, (**b**) Full wrist flexion on both sides, (**c**) Marginal supination, and (**d**) Pronation deficit on the right compared to the left side.

**Table 1 jcm-12-04981-t001:** Postoperative Outcome after FVFGP transfer.

Author	Year	Ref.	Nr of Cases	Age	Neuro-Fibromatosis	Follow-Up	Complications	Further Surgeries	Union
Mathoulin	1993	[6]	1	3 y	Yes	144 mt	None	None	Yes
Masterson	1993	[14]	2	5 y	Yes	9 mt	None	None	Yes
Cheng	1994	[15]	3	9 mt	Yes	8 mt	None	None	Yes
Allieu	1999	[12]	4 5	22 y 17 y	n/a n/a	17 y 13 y	Yes Yes	None None	Yes Yes
Witoonchart	1999	[10]	6	1 y	No	48 mt	None	None	Yes
			7	5 y	Yes	24 mt	Yes	Yes	Yes
Suzuki	2005	[17]	8	4 y	n/a	96 mt	Yes	Yes	Yes
			9	4 y	n/a	96 mt	Yes	Yes	Yes
			10	1 y 7 mt	n/a	60 mt	None	Yes	Yes
Bae	2005	[1]	11	15 y	No	81 mt	Yes	Yes	Yes
			12	3 y	Yes	78 mt	Yes	Yes	Yes
			13	5 y 3 mt	Yes	31 mt	None	None	Yes
			14	16 y	Yes	42 mt	Yes	Yes	Yes
El Hage	2009	[18]	15	18 mt	Yes	72 mt	Yes	Yes	Yes
			16	12 y	Yes	36 mt	Yes	Yes	Yes
Bauer	2013	[7]	17	9 y 6 mt	Yes	31 mt (range 23 to 83 mt)	Yes	Yes	Yes
			18	2 y 8 mt	Yes		None	None	Yes
			19	11 y 6 mt	n/a		None	None	Yes
			20	7 y	No		Yes	None	Yes
			21	12 y 6 mt	Yes		None	None	Yes

**Table 2 jcm-12-04981-t002:** Postoperative ROM in included case reports.

Author	Ref.	Nr of Case	ROM Wrist/Elbow
Mathoulin	[6]	1	wrist: normal, elbow: normal, P/S: 90/0/50°
Masterson	[14]	2	wrist: full flexion, extension 25°, elbow: full ROM
			P/S: 30/0/90°
Cheng	[15]	3	wrist: F/E: 60/0/30°, elbow: n/a, P/S: 60/0/50°
Allieu	[12]	4	wrist: F/E: 80/0/80°, elbow: normal function, P/S: 30/0/80°
		5	wrist: F/E: 70/0/5°, elbow: n/a, P/S: Neutral position
Witoonchart	[10]	6	wrist: n/a, elbow: n/a, P/S: 10/0/80°
		7	wrist: normal function, elbow: normal function
			P/S: 10/080°
Suzuki	[17]	8	wrist: F/E: 25/0/45°, elbow:F/E: 130/0/0°, P/S: 20/0/60°
		9	wrist: F/E: 60/0/65°, elbow: F/E 120/5/0°, P/S: 35/0/75°
		10	wrist: F/E: 55/0/60°, elbow: F/E 125/10/0°, P/S: 80/0/80°
Bae	[1]	11–14	n/a
El Hage	[18]	15	wrist: n/a, elbow: F/E 130/0/0, P/S: 70/0/70
		16	wrist: n/a, elbow: F/E 120/0/0, P/S: 70/0/60°
Bauer		17	normal function
		18	normal function
		19	normal function
		20	normal function
		21	P/S:45/0/85°

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
