# Peer review of "Vascularized Growth Plate Transfer in Paediatric Ulna Non-Union: Operative Technique and Review of the Literature"

_jcm, 2023, doi:10.3390/jcm12154981_

Round 1
Reviewer 1 Report
Interesting paper on rare condition and technique. Some modifications are needed:
Time frame of Figure 1 and 2 is not completely clear to me; X-ray just before osteotomy would be nice, as X-ray at 2 years after osteotomy (same time of Figure 6). There is no Figure 4. Figure 3 is not reported in the text (maybe somewhere around line 135)
Table “1” (the table is missing number and title): Why Allies is reported at the end (I belie you used a chronological order for other citations), and number of cases does not appear correctly for Allieu. Furthermore, from table 1 it seems you included 21 cases in line 184 you report 20, 22 in line 186… Then 186-187 you say “Ten of these patients… For the remaining nine”???
Can’t find the Allieu paper of 1999 in the references
In the literature review when available it would be nice to also report the motions obtained at latest follow-up.
I belve that nothing you wrote in your review demonstrated that “FVFGP transfers are mainly used to reach 246 the following three goals: 1. bony healing, 2. articular stability and 3. preservation of 247 growth”, please edit line 246-248.
Line 251 “This in theory should allow for better bone healing. “ add reference
Some minor typing/English mistakes have to be checked, eg:
- Line 53 you are missing a period before “Charles”
- Line 221 Elfattairy et at.
- Review sentence in line 224-226
- Line 265-266 change entrance (maybe just erase the final “is”)
- Etc…
Author Response
Interesting paper on rare condition and technique.
Thank you for your time and effort enhancing our paper.
Some modifications are needed:
Time frame of Figure 1 and 2 is not completely clear to me;
Thank you – we have added information about when these xRays were taken.
X-ray just before osteotomy would be nice, as X-ray at 2 years after osteotomy (same time of Figure 6).
I’ve added the xRays at your wish
There is no Figure 4. Figure 3 is not reported in the text (maybe somewhere around line 135)
Thanks for noticing – we have updated the numeration
Table “1” (the table is missing number and title): Why Allies is reported at the end (I belie you used a chronological order for other citations), and number of cases does not appear correctly for Allieu.
Thank you for noticing – we’ve renumbered the table and changed the order according publication date and name. Due to some editing changes the numeration was confused. We have corrected this issue.
Furthermore, from table 1 it seems you included 21 cases in line 184 you report 20, 22 in line 186… Then 186-187 you say “Ten of these patients… For the remaining nine”???
There have been 21 cases so far - plus ours = 22. I’ve updated the numbers and hopefully made it clearer in the results section which cases have been added.
Can’t find the Allieu paper of 1999 in the references
We have added this reference
In the literature review when available it would be nice to also report the motions obtained at latest follow-up.
Thank you for your interest. I’ve added a table on this. The evolution of the technique is such that nowadays motion is much better retained.
I belve that nothing you wrote in your review demonstrated that “FVFGP transfers are mainly used to reach 246 the following three goals: 1. bony healing, 2. articular stability and 3. preservation of 247 growth”, please edit line 246-248.
This is a quote from the articles that have been published. We have added the references from the references. Indications were not discussed in the literature review.
“Our review of the literature demonstrated FVFGP transfers are mainly used to reach the following three goals: 1. bony healing, 2. articular stability and 3. preservation of growth [4, 16, 18, 19, 26]. In comparison to classical cancellous bone graft techniques, it provides two important advantages: it allows for a reconstruction of long segmental bone defects, which means that a more radical debridement of the pseudoarthrosis is permitted [27]. This in theory should allow for better bone healing in these challenging cases, which was confirmed in our series with a primary healing rate of 54%.
Line 251 “This in theory should allow for better bone healing. “ add reference
This is a conclusion from our review and case – see above.
Comments on the Quality of English Language
Thanks – these have been changed and also the structure of the article was adjusted to the needs of the journal.
Some minor typing/English mistakes have to be checked, eg:
- Line 53 you are missing a period before “Charles”
- Line 221 Elfattairy et at.
- Review sentence in line 224-226
- Line 265-266 change entrance (maybe just erase the final “is”)
- Etc…

Reviewer 2 Report
The Authors presented a paper regarding a very rare condition (ulnar pseudoarthrosis) and reported their experience with the vascularized growth plate transfer for it.
The paper is well written and provide the correct informations and citations to look for the procedure.
I agree with the Authors that, considering the scarcity of cases, it's very difficult to achieve statistical evidence regarding this kind of treatment. Furthermore, I appreciate the Literature review in the paper.
I'd like to address just some minor revision:
-Line 19: you mispelled PubMed ("Pupmed")
-Line 183-186: regarding the number of cases, you first stated 20 cases, then you reported final osseus consolidation for 22 patients, finally in the table you reported 21 cases (there is also a typo in the last line of the Table for Allieu's paper). There is clearly a confusion about it. Please correct and modify
-Also, mispelled in the table for Allieu "reported" ("reporetd")
Author Response
The Authors presented a paper regarding a very rare condition (ulnar pseudoarthrosis) and reported their experience with the vascularized growth plate transfer for it.
The paper is well written and provide the correct informations and citations to look for the procedure.
I agree with the Authors that, considering the scarcity of cases, it's very difficult to achieve statistical evidence regarding this kind of treatment. Furthermore, I appreciate the Literature review in the paper.
Thank you for your efforts and time to help us enhancing the paper. For the structural needs of the journal we had to change some aspects. We hope that you still like the literature review which is now included in the results section and discussion.
I'd like to address just some minor revision:
- -Line 19: you mispelled PubMed ("Pupmed")
Thank you – this has been changed.
- -Line 183-186: regarding the number of cases, you first stated 20 cases, then you reported final osseus consolidation for 22 patients, finally in the table you reported 21 cases (there is also a typo in the last line of the Table for Allieu's paper). There is clearly a confusion about it. Please correct and modify
Thanks we have updated these issues. These were due to editing changes and are now corrected. And the order of papers is now by date of publication and name of primary author
- -Also, mispelled in the table for Allieu "reported" ("reporetd")
Thank you , this has been changed.
